# Drug Discovery in the Age of Artificial Intelligence: Transformative Target-Based Approaches

**DOI:** 10.3390/ijms252212233

**Published:** 2024-11-14

**Authors:** Akshata Yashwant Patne, Sai Madhav Dhulipala, William Lawless, Satya Prakash, Shyam S. Mohapatra, Subhra Mohapatra

**Affiliations:** 1Center for Research and Education in Nanobioengineering, Department of Internal Medicine, Morsani College of Medicine, University of South Florida, Tampa, FL 33612, USA; apatne@usf.edu; 2Taneja College of Pharmacy Graduate Programs, MDC30, 12908 USF Health Drive, Tampa, FL 33612, USA; 3Department of Molecular Medicine, Morsani College of Medicine, University of South Florida, Tampa, FL 33612, USA; saimadhavdhulipala@usf.edu (S.M.D.); wlawless@usf.edu (W.L.); 4Research Service, James A. Haley Veterans Hospital, Tampa, FL 33612, USA; 5Biomedical Technology and Cell Therapy Research Laboratory, Department of Biomedical Engineering, Faculty of Medicine and Health Sciences, McGill University, 3775 University Street, Montreal, QC H3A 2B4, Canada; satya.prakash@mcgill.ca

**Keywords:** drug discovery, graph neural networks, random forests, general adversarial networks, target-based approaches, phenotypic approaches

## Abstract

The complexities inherent in drug development are multi-faceted and often hamper accuracy, speed and efficiency, thereby limiting success. This review explores how recent developments in machine learning (ML) are significantly impacting target-based drug discovery, particularly in small-molecule approaches. The Simplified Molecular Input Line Entry System (SMILES), which translates a chemical compound’s three-dimensional structure into a string of symbols, is now widely used in drug design, mining, and repurposing. Utilizing ML and natural language processing techniques, SMILES has revolutionized lead identification, high-throughput screening and virtual screening. ML models enhance the accuracy of predicting binding affinity and selectivity, reducing the need for extensive experimental screening. Additionally, deep learning, with its strengths in analyzing spatial and sequential data through convolutional neural networks (CNNs) and recurrent neural networks (RNNs), shows promise for virtual screening, target identification, and de novo drug design. Fragment-based approaches also benefit from ML algorithms and techniques like generative adversarial networks (GANs), which predict fragment properties and binding affinities, aiding in hit selection and design optimization. Structure-based drug design, which relies on high-resolution protein structures, leverages ML models for accurate predictions of binding interactions. While challenges such as interpretability and data quality remain, ML’s transformative impact accelerates target-based drug discovery, increasing efficiency and innovation. Its potential to deliver new and improved treatments for various diseases is significant.

## 1. Introduction

For centuries, the quest to discover life-saving medications has been a relentless pursuit fraught with challenges and uncertainties. The intricate journey from identifying a disease target to delivering a safe and effective drug remains a marathon, often hampered by limited speed, efficiency, and success. However, recent advancements in artificial intelligence (AI) have ignited a spark of hope, injecting a transformative force into the drug discovery landscape.

The complexities that are inherent in drug development are multi-faceted. Targets, often intricate proteins or enzymes, may harbor hidden mechanisms or allosteric sites, making intervention difficult. The vast chemical space, with an astronomical number of potential molecules, poses a daunting challenge when it comes to identifying the right drug candidate. Ensuring drug safety, efficacy, and affordability also requires navigating regulatory hurdles. These unmet needs have fueled the search for innovative solutions, and AI has emerged as a powerful tool to revolutionize the drug discovery process.

Numerous reviews have explored the potential of AI in drug discovery, focusing on specific aspects like virtual screening, target identification, or deep learning applications. While valuable, these reviews often lack a comprehensive overview encompassing the diverse range of AI-driven approaches within target-based and phenotypic strategies.

This review aims to bridge this gap by offering a holistic perspective on the transformative impact of AI in drug discovery. We delve into the intricacies of target-based approaches, exploring advancements in small-molecule, fragment-based, and structure-based methods. We then shed light on the potential of phenotypic approaches, leveraging AI to analyze cell-based assays and genetic screens. Beyond these specific strategies, we examine the broader contributions of AI in drug repurposing, computational predictions, and personalized medicine.

## 2. Methods

Machine learning (ML) is an emerging field derived from AI, an idea that began with Alan Turing in the 1940s and has accompanied the computer revolution over the last four decades. The discovery of carbon nanomaterials, including carbon nanotubes and graphene, in 2004 provided impetus for early AI algorithms for the game of checkers, facial image recognition, and self-driving cars [1]. In short, ML encompasses the study and training of various types of algorithms that use input from datasets to predict an output independently [2]. Through several trials, a user improves the set of algorithms, known as a model, to make more accurate predictions [3]. The intricate complexities of disease targets and the vast chemical landscape have long burdened the relentless pursuit of effective medications. Traditional target-based approaches, while revolutionizing drug discovery, have also faced limitations. High-throughput screening (HTS) offered rapid lead identification, but false positives and neglect of drug-like properties hampered progress. Fragment-based and structure-based approaches and virtual screening (VS) provided finesse, but challenges remained. Fortunately, the dawn of AI involving ML and deep learning approaches has ignited a transformative era, empowering each approach with unprecedented capabilities.

In the context of ML, the field can be broadly categorized into supervised and unsupervised learning. Supervised learning involves training algorithms on labeled data to predict new, unseen data, while unsupervised learning identifies patterns and structures within unlabeled data. Common examples of supervised learning tasks include classification and regression, while clustering and dimensionality reduction are typical unsupervised learning tasks, as shown in Figure 1.

### 2.1. Small Molecule-Based Approach (SMA)

The traditional workhorse of target-based drug discovery was HTS. This approach initially involved physical testing of vast libraries of small molecules against specific biological targets, like proteins or enzymes, to identify potential lead candidates. These physical assays were later automated by machine pipetting and computer tracking, which increased speed and validation while reducing labor and material costs. While HTS revolutionized drug discovery by accelerating lead identification, it faced significant limitations. Many screened compounds often resulted into higher false-positive rates, requiring further validation and optimization. Additionally, HTS predominantly focused on ligand binding without considering crucial aspects like drug-like properties, pharmacokinetics, and toxicity, potentially leading to failures later in development [6].

These limitations were addressed with the development of computational-based VS methods, which precede experimental verification. VS is performed on thousands or even millions of compounds to create a top-ranking small-molecule interaction derived from physics-based computational calculations that measure predicted binding free energy. Analysis can be performed as either a structure-based drug design, which employs the biomolecular structure, or as a ligand-based design, which does not require a structure. VS also faces several challenges, such as the need for user knowledge about the binding target structure to avoid high computational costs, increase binding accuracy, and avoid erroneous assumptions. The computational cost in terms of both time and machine investment can be high, and calculations based on poorly established coordinates for binding pockets or overly large binding pockets can increase the computational time exponentially.

ML solves several problems associated with traditional VS and can augment structure- and ligand-based drug design with remarkable accuracy. ML incorporates training data into an analytical method and utilizes a separate set of validation data that assess a given model’s prediction accuracy and precision, determining the best possible ML model for a specific demand. Figure 2 elaborates on the small molecule-based approaches divided into supervised and unsupervised learning and various models, methods, input, and output using ML.

Recent advancements in ML have ignited a transformative era in target-based drug discovery [7]. ML models, empowered with vast datasets of chemical and biological information developed originally for VS, can predict small molecules’ potential binding affinity and selectivity with remarkable accuracy without any initial physical assay, dramatically reducing the number of compounds that require experimental validation. Furthermore, generative models can design novel small molecules with desirable properties, expanding the search space for promising lead candidates [8].

**Figure 2 ijms-25-12233-f002:**
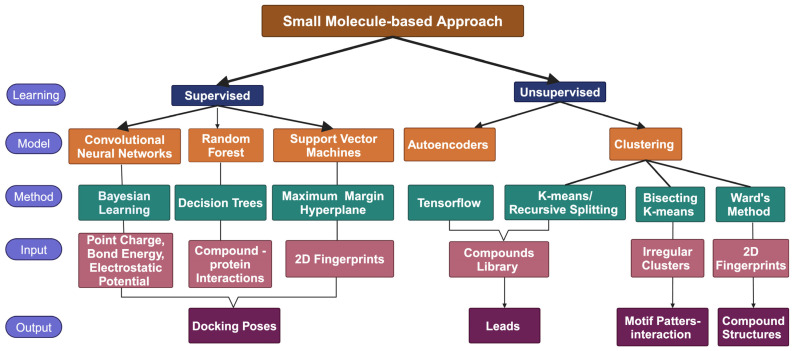
Example of algorithms and classifiers in ML models for small molecule-based approach drug discovery [9].

#### 2.1.1. SMA-Supervised Learning

Supervised learning algorithms leverage established knowledge (also called test data) to make predictions. In supervised methods, we can use a controlled approach with large datasets to develop confidence in known relationships or work on meta-relationships to identify leads from big datasets. Some prominent examples used in the small-molecule approach are listed below.

#### 2.1.2. SMA-Support Vector Machines (SVMs)

This model finds the optimal hyperplane, a decision boundary used to maximize the margin between different data points, which helps filter the noise of the data and increase prediction accuracy. Interdependent training sets—including random, closely related, highly active, and boundary molecules—can be used to improve small molecule-binding predictions. The project stage at which ML models are applied dictates the choice of training set. Some of the examples of SVM models and their applications are shown in Table 1. One study investigated the difference between active and passive training methods and showed that the iterative selection of the model improved hit prediction accuracy by 29% for the thrombin dataset. This study also observed the advantages of exploitation using only the highest positive scores over exploration near a decision boundary for fewer data points, and the near equal efficiency of both when there are more data points [10]. Meanwhile, some studies used SVMs for predicting and scoring optimal docking poses. These studies showed predictions of consistent RMSD values (>0.9) and low prediction error (0.25), indicating high predictivity with minimal overfitting [11]. Studies working with novel protein targets without known 3D structures trained the vector machine models on 2D fingerprints of the chemical structure of a protein sequence. They utilized the linear combination method to yield a significant improvement in the prediction of compound activity compared to homology modeling-based predictions, as evidenced by the recovery rates of the projections. SVMs built using the linear combination method showed a 50% improvement in recovery rates for all target molecules [12].

#### 2.1.3. Random Forests (RFs)

In this model, random subsets of data are organized as nodes, and these subsets are used to “grow” the tree. A prediction is made by aggregating all these trees. These models have proven to be very efficient with docking pose predictions [15]. The Matthews Correlation Coefficient (MCC) has been used to evaluate prediction model performance by considering true and false positives and negatives [16]. Higher MCC values, indicating better predictions, were achieved, with models reaching MCC values of 0.8 within 2000 to 3000 iterations and 0.6 within 3000 iterations. RF implementations combine the output of decision trees to handle classification and regression problems, predicting IC50 values of drug interactions and demonstrating statistical significance in two-tailed *t*-tests [17]. RDKit generates 2D depictions in PDBe CCDUtils, aiding RF regression to predict drug sensitivity with high accuracy [18]. These examples show the relevance and advantage of RF methods, which, even with minimal parameter tuning, demonstrate faster learning and balanced selection strategies when dealing with high-scale multivariate data.

#### 2.1.4. Convolutional Neural Networks (CNNs)

These models were designed to use image data. They use some layers to extract features from images and some layers to reduce dimensionality. Then, all layers are combined to make the final predictions. These models work best in applications with minimal feature engineering and image data. They have been used in studies to analyze graphs built upon preliminary sequential data or fingerprints [19]. In pharmacokinetics, the Area Under the Curve (AUC) refers to the definite integral of the concentration of a drug in blood serum as a function of time, serving as an essential predictor of drug bioavailability. One study explored CNN architectures for classifying ligands of cannabinoid receptors, achieving impressive results with AUC values ranging from 0.693 to 0.944 across different datasets, with the LeNet-5 architecture consistently outperforming others, as demonstrated by AUC scores peaking at 0.942 for AtomPair fingerprints on the CB1 test set [20]. Another study predicts electrostatic potential (ESP) surfaces for proteins and ligands using graph-convolutional deep neural network (DNN) models. ESP maps account for the overall strength of adjacent charges to a given point within a molecule and can predict molecular interactions with target residues within binding pockets. Trained against density functional theory (DFT) ESP surfaces, the ligand deep neural network fingerprint (DNN-fp) model outperforms AM1-BCC, providing fast, high-quality predictions that correlate strongly with experimental molecular properties, enabling interactive drug design [21]. Currently, the biggest challenges associated with supervised learning methods are the requirement of larger datasets and the possibility of bias in parameter selection in various approaches.

#### 2.1.5. SMA-Unsupervised Learning

Unsupervised learning algorithms can draw patterns within data without predetermined labels. This makes them particularly valuable for exploring novel chemical spaces and identifying promising lead candidates for targets without known actives. However, unsupervised learning can also contribute significantly to targets with known binders. In contrast to supervised methods, these frameworks require a smaller database and are more suitable for establishing new relationships and hidden patterns between parameters.

##### Clustering Algorithms

Clustering is suitable for identifying patterns by grouping similar data naturally without drastically affecting the data size [22]. This method is particularly useful when dealing with incomplete input data, such as parts of input sequences or sections of structural fingerprint ensembles, where parts of the dataset are unavailable [23]. There are three prominent frameworks for clustering models: K-means, bisecting k-means, and Ward’s algorithm. K-means has been applied in deep clustering methods for proteins with no known active agents. By identifying leads for target proteins based on similarity and structure–activity relationships, k-means has shown impressive accuracy in predicting potential targets [24]. For example, in some applications, clusters have successfully predicted the binding affinity of protein–ligand interactions, making it a valuable tool in drug discovery. Bisecting k-means improves upon traditional k-means by recursively splitting lower-level clusters into subclusters [25]. This approach is particularly effective for irregular datasets, outperforming randomized k-means initialization by detecting significant motif patterns. For instance, bisecting k-means has been used to identify rare but important structural motifs in protein–ligand binding sites, improving the precision of lead identification over standard k-means methods. Ward’s method, which is commonly employed for quantitative variables when binary variables do not exist, minimizes variance within clusters. It excels in applications with consistent compound families, eventually form a single cluster. Ward’s method has been utilized to implement the Székely–Rizzo clustering approach to determine compound structures based on 2D fingerprints [26]. This has been instrumental in grouping compounds with similar chemical properties, and aiding the design of novel compounds in drug discovery.

##### Autoencoders

These models use an encoder to compresses the input data and a decoder to reconstructs new data from the compressed data. They have a high prediction accuracy, particularly when dealing with large datasets [27]. For example, TensorFlow-based autoencoders have improved predictions and overall accuracy in the context of the breast cancer gene GL50, spurring the development of comprehensive datasets for other ML models used in drug discovery [28].

Another study utilizes SMILES structures from the CHEMBL database to prepare a low-dimensional latent space representation and generate topographic maps. These maps could be used to prepare a compound library with insights into accessibility based on potential synthesis process complexity and latent descriptors. Data-driven latent vectors provided a much better representation of the compounds due to flexibility with input data, bringing nuance to the library [29].

One study displayed a direct advantage of unsupervised methods during pre-training, as demonstrated by very high prediction accuracy when a restricted Boltzmann machine-based implementation of a deep belief network built on key fingerprints from the molecular access system (MACCS) [30].

However, the source of those hidden relationships from the above-mentioned unsupervised models can be difficult to deduce, as they are built upon themselves with low-dimensionality input data. This can lead some fundamental supervised methods to outperform the unsupervised methods downstream.

### 2.2. Fragment-Based Approach (FA)

The fragment-based approach differs from the small-molecule approach by utilizing partial segments of pre-established structures. This is more useful for understanding structure–activity relationship studies [22]. The smaller fragments simplify the optimization of lead compounds, exponentially expediting the drug discovery process by eliminating the need to calculate all chemical interactions using various methods, as described in Figure 3 [23].

AI plays a similar role to the previous approach in advancing fragment-based drug discovery, contributing to various tasks crucial for optimizing lead compounds. The ML algorithms utilized in this context are significantly different, with projects using AI in a fragment-based approach have a broader-range of parameter selection—Such as binding affinity, energy, or entropy.

#### 2.2.1. FA-Supervised Learning

##### Convolutional Neural Networks

These algorithms have been used to analyze the relationship between fragment data and binding pocket interactions by predicting fragment structure based on the SMILES format sequence or by including characteristics like chemical properties [25]. One study factored in the free energy of the fragments to predict docking poses [26]. This technique has been used to form a shrinkage logistic regression model to predict the fragment interaction [31]. Some frameworks of neural networks have been modeled based on the Monte Carlo dropout method, trained mostly on the binding affinity and binding energy of different compounds to estimate uncertainty and improve predictability. One study employed molecular dynamics (MD) simulations to predict the free energies of 15,000 small molecules transferred between water and cyclohexane using a 3D-CNN. The results demonstrate the prediction of 2.5–5 KJ/Mol, aligning with experimental models [32].

#### 2.2.2. FA-Support Vector Machines (SVMs)

As stated in Shahab et al.’s study which predicted binding pockets and fragment properties, SVMs can classify and analyze large datasets of fragments to identify those with specific characteristics relevant to drug development. For instance, they have been used to predict the binding modes of kinase inhibitors based on X-ray structures as templates and have proved to be a reliable method for building large libraries [33]. It is important to note that SVMs were suggested to be more suitable for downstream validation or precision predictions of structures over preliminary fingerprint-based predictions until selectivity parameters like IC50 were utilized during training, yielding 92% prediction accuracy [34].

#### 2.2.3. Reinforcement Learning

Reinforcement learning is more suitable for iterative scoring in training stages of the model; the greater the number of iterations, the better the model’s scoring performance and results [35]. One recent study utilized molecular graph transformers based on compounds from CHMEBL and LIGAND databases [36]. The researchers iteratively reinforced the model with scoring based on interactive parameters of small fragments, such as drugs’ similarities to and affinity towards their study target protein A2AAR, to generate structures using the SMILES format, with an exploitation-focused approach [37]. These models could generate fragments that were highly compatible with A2AAR [38]. Another study explored structural characteristics like fragment length, branching, and bond flexibility. This study focused on the scoring strategy based completely on the raw potential of fragments. The scoring for this model focused more on exploration. The generated fragment libraries have a range just as vast as the input libraries [39]. Thus, this approach would be more advantageous for compound synthesis than the case-/disease-specific transformer application.

#### 2.2.4. FA-Unsupervised Learning

##### Clustering

Studies show high prediction accuracy for the construction of molecular fingerprints based on two Word2vec models (with skip-gram (SG) and continuous bag of words (CBOW) implementation), developing t-distributed stochastic neighbor embedding (t-SNE) plots and QSAR modeling [40]. Specifically, kinase inhibitors and anti-HIV compounds showed sensitivity of 77% and specificity of 87% for distributed fingerprints, with sensitivity of 67% and specificity of 91% for fragment fingerprints [41]. The clustering approach shines in these studies, achieving high-accuracy predictions despite the lack of labeled data [39]. It shows a demonstrable advantage over conventional QSAR modeling approaches using supervised models like CNNs. It might also be better than other unsupervised approaches like autoencoders due to their efficiency in capturing complex relationships without encoding or decoding, which might be more relevant to the small-molecule approach [40].

### 2.3. Structure-Based Approach (SA)

The structure-based approach leverages high-resolution protein structures, allowing scientists to design ligands with exquisite specificity. Recent advancements in ML are further empowering this technique, particularly with regard to challenging targets for which crystal structures are elusive; some of the relevant techniques are mentioned in Figure 4 [42]. Predictions based on previous templates from various databases using homology modeling [43], threading [44], or ab initio [45] prediction of folding confirmations have proved to be intuitive applications for ML in a structure-based approach.

#### 2.3.1. SA-Supervised Learning

##### Generative Adversarial Networks (GANs)

The ProteinGAN model, trained on a diverse dataset of 16,706 unique sequences of bacterial malate dehydrogenase enzymes, yielded significant predictive outcomes. It achieved a median sequence identity of 61.3% for natural sequences and identified 119 novel structural motifs [47]. Another study used GAN with spectral normalization to achieve tight backbone distribution of the sequences and stability [48]. GANs significantly increased the accuracy of protein folding predictions, helping researchers identify binding pocket more accurately and select suitable ligands for compounds that bind with kinase and dopamine receptors [49].

#### 2.3.2. SA-Support Vector Machines

The SWISS-MODEL uses multiple sequence alignments, interface geometry, and residue entropy distribution to identify templates that maximize high-quality estimates based on inter-chain contact scores [50]. The model uses Monte Carlo sampling with Pro-Mod3 [51], which employs a library of parameters for energy minimization. Each developed model is scored based on its evolutionary significance [52]. SVMs can find optimal decision boundaries in high-dimensional feature spaces (here, interface conservation and geometric properties), effectively separating suitable templates from unsuitable ones.

#### 2.3.3. Structure Networks

Sequence alignment Networks have been used to establish evolutionary relationships, and an encoding module is trained based on previous structures to predict contacts in amino acids and, as a result, their folding pattern [53]. These networks are particularly useful here, as they can identify long-range dependencies [54], so they can identify amino acid interactions by using the attention mechanism [55] with local alignments when electron densities and ion spheres are factored into the implementation of these models’ parallel processing. The attention modules showed a Pearson coefficient of 0.78 with precision scores for all targets, demonstrating their relevance to larger and more complex databases without the high data requirements of GANs or feature engineering required by SVMs.

#### 2.3.4. SA-Unsupervised Learning

##### Protein Language Models (PLMs)

Some projects, like Omegafold, which applies PLMs based mostly on unaligned and unlabeled sequences [56], use residue pairs to create embeddings [57]. These embeddings are refined by geometric consistency, mostly to improve distance predictions [58]. One study used the harmonic mean of true positives and recall to observe the class distributions (F1 score) of predictions of molecular function, biological process, and cellular component-specific relation with the sequence using two protein language models: K-sep and SeqVec. Both models performed well in terms of molecular function prediction, with F1 scores of 916 and 914, respectively [59]. This approach is significantly different from that of Alphafold2, where predictions depend less on comparative parameters like evolutionary significance or deviation and more on spontaneous metadata derived directly from the sequence, outperforming Alphafold2 when model training data do not extend beyond the sequence [60].

##### Clustering

In a novel decoy selection-based clustering model, computer-developed decoys are used to organize decoys and original structures to identify noise. This is implemented using a k-means framework in which decoy region identification receives greater focus than the folding confirmation selection, which makes this type of model very flexible as it can be used as a refinement method for structures predicted by other models [61].

### 2.4. Examples of ML Affecting Bioinformatics Drug Discovery

ML has significantly advanced the field of bioinformatics, a major facet of bio-nanotechnology involving the interaction of nanomaterials with biological systems comprising DNA, RNA, proteins, and metabolism, by enabling the analysis of complex biological data in previously impossible ways. Some key areas where machine learning has had a substantial impact include the fields of genomics, proteomics, transcriptomics, systems biology, and drug discovery. Some examples of ML’s contributions to bioinformatics are shown in Table 2.

ML algorithms can analyze vast amounts of genomic data to identify patterns and mutations associated with diseases. This helps further our understanding of genetic predispositions and assists with the development personalized medicine [71]. Additionally, ML techniques help us to analyze RNA sequencing data to understand gene expression patterns, which is vital for studying how genes are regulated and how they respond to different conditions [72]. By integrating data from various biological sources, ML aids in complex modeling scenarios.

Traditional methods of protein network analysis can be time-consuming and computationally intensive. ML algorithms can process large datasets more quickly, helping us understand the behavior of biological systems under different conditions [73]. By analyzing protein structures and functions, ML aids in the prediction of protein interactions and functions, which is crucial for drug discovery and understanding cellular processes [74]. This efficiency makes it feasible to analyze complex networks on a larger scale [75]. Further, ML can model the dynamic behavior of protein networks under different conditions, such as changes in the environment or disease states, helping us understand how protein interactions change over time [76]. These advancements have accelerated research and enhanced our understanding of cellular processes, leading to potential breakthroughs in drug discovery and personalized medicine.

ML algorithms have been used to reconstruct metabolic networks by integrating various types of omics data (e.g., genomics, transcriptomics, and proteomics). ML has significantly improved the efficiency of protein network analyses in several ways. First, ML models can predict interactions between proteins by analyzing large datasets of known interactions, aiding in the construction of more accurate and comprehensive protein interaction networks [77]. Second, by analyzing patterns in protein sequences and structures, ML can predict the functions of unknown proteins, aiding in the annotation of protein networks [78]. Additionally, ML algorithms can integrate various types of biological data (e.g., genomic, transcriptomic, and proteomic data) to reconstruct protein interaction networks, providing a more holistic view of cellular processes [79]. Thus, ML models can predict how different compounds will interact with biological targets, speeding up the drug discovery process and reducing costs [80]. These advancements have not only accelerated research but have also opened new avenues for personalized medicine, making treatments more effective and tailored to individual patients.

Despite the progress made in bioinformatics, applying ML to biology comes with several challenges. First and foremost, biological systems are incredibly complex and dynamic [81] making it difficult to create accurate models that can predict biological behavior. Successful ML application requires high-quality, annotated datasets that are essential for training ML models. However, biological data often contain noise, missing values, and inconsistencies, which can hinder a model’s performance. Additionally, the effective application of ML in biology requires expertise in both fields. Bridging the gap between computational scientists and biologists can be challenging due to differences in terminology, methodologies, and objectives [82]. Furthermore, biological datasets can be enormous, requiring significant computational resources for processing and analysis. Ensuring that ML models can handle these large datasets is a major challenge [83]. Another challenge is that many ML models, especially deep learning models, are often seen as “black boxes”. Understanding how these models make predictions is crucial for gaining biological insights and ensuring trust in the results. Finally, handling sensitive biological and medical data raises ethical and privacy issues. Ensuring data security and patient confidentiality is paramount [84]. Despite these challenges, the potential benefits of applying ML to biology are immense, driving ongoing research and innovation in this exciting field.

## 3. Conclusions

The integration of ML into target-based drug discovery represents a monumental leap in the pharmaceutical industry. By enhancing traditional methods such as HTS and VS, ML models have significantly improved the prediction accuracy of binding affinities and selectivity, thereby reducing the need for extensive experimental screening. The application of deep learning techniques, such as CNNs and RNNs, shows great promise in virtual screening, target identification, and de novo drug design. Additionally, fragment-based and structure-based approaches have benefited from ML algorithms that predict fragment properties and binding affinities with remarkable precision.

The advent of GANs and other advanced techniques has further empowered researchers to explore and expand the chemical space, enabling the discovery of novel molecules with desired properties. Despite challenges such as interpretability and data quality, the transformative impact of ML is undeniable, accelerating the drug discovery process and fostering innovation.

Given the widespread use of computational algorithms in predicting experimental protein structures and the increasing reliance on virtual screening for lead selection, it is conceivable that the efficiency of computational methods will eventually rival or surpass traditional experimental methods in terms of resolution and accuracy. This could potentially address the limitations inherent in each of these approaches.

In summary, the application of ML in target-based drug discovery is paving the way for more efficient and effective identification of therapeutic candidates and significantly improves upon methods traditionally established by HTS and VS. As ML models continue to evolve, they have the potential to revolutionize the development of new and improved treatments for various diseases, enhancing patient outcomes and advancing medicine.

## Figures and Tables

**Figure 1 ijms-25-12233-f001:**
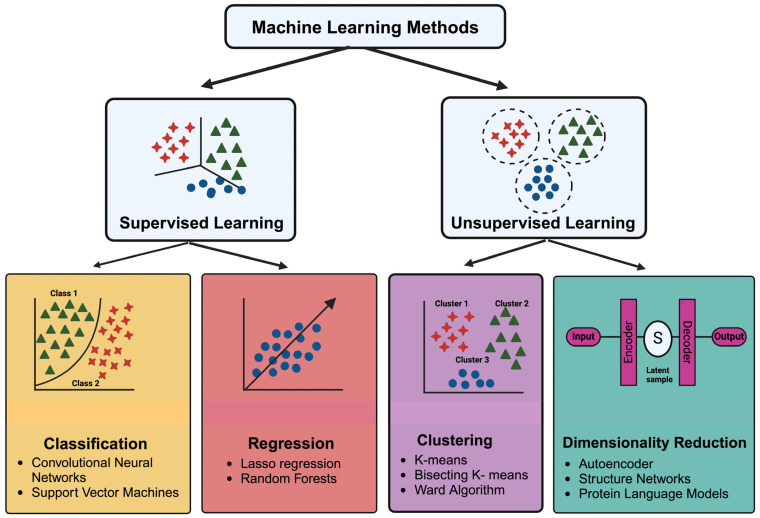
Example of algorithms and classifiers in ML models [4,5].

**Figure 3 ijms-25-12233-f003:**
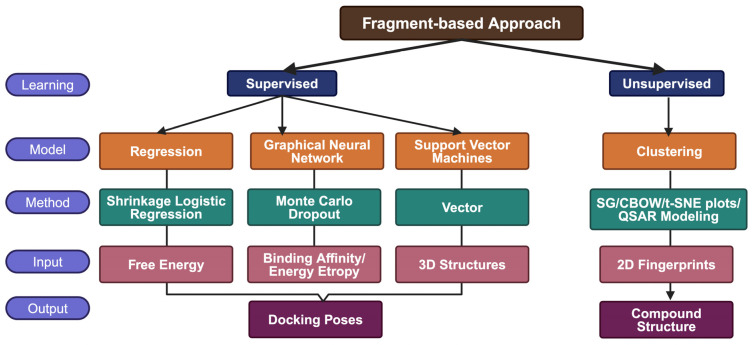
Example of algorithms and classifiers in ML models for a fragment-based approach to drug discovery [24].

**Figure 4 ijms-25-12233-f004:**
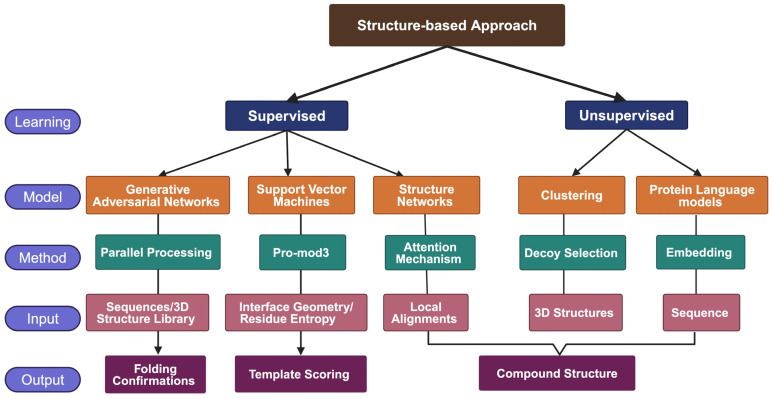
Example of Algorithms and Classifiers in ML Models for Structure-Based Approach Drug Discovery [46].

**Table 1 ijms-25-12233-t001:** A select list of SVM models and applications.

Application	SVM Advantage	References
Predicting activity based on 2D chemical structures.	Penalty cost functions help with data point prioritization based on certainty, reduces reruns.	[10]
Predicting binding affinities based on 3D chemical structures.	Kernel SVMS transform non-linearly bound data can be used to produce linear relationships.	[11]
Predicting compatibility of ligands based on protein sequences.	Pattern recognition with limited information.	[12]
Predicting activity based on 3D chemical structures.	High-dimensional data classification.	[13]
Predicting drug-to-drug interactions based on structural similarities.	Drug pair identification and classification.	[14]

**Table 2 ijms-25-12233-t002:** Examples of ML’s role in drug discovery by increasing the speed and efficacy of bioinformatics.

Field	Program/ML-Technique	Benefits	Application
Genomics	Anomaly detection using unsupervised learning speeds up the identification of disease-associated genes but also improves the accuracy of predictions.	Quickly identifies genetic mutations and variations across large datasets	Personalized medicine and targeted therapies [62]
	CellProfiler [63]	Conducts the automatic analysis of biological images	It helps detect subtle changes and patterns in cells
Transcriptomics	Sc RNAseq [64]: Clustering and dimensionality reduction (e.g., t-SNE, UMAP) allow researchers to quickly identify and visualize distinct cell populations within complex datasets	Accelerates the discovery of new cell types and states, enhancing the understanding of cellular diversity and function	Useful in disease diagnosis and therapy
	Spatial transcriptomics [65] involves deep learning models to analyze data, identify spatially variable genes, and reconstruct spatial gene expression patterns	Improves the resolution and accuracy of spatial maps	Provides insights into tissue organization and mechanisms and develops therapies
Proteomics	Percolator: Semi-supervised rescoring of peptide-spectrum matches (PSMs) [66]	Significantly boosts the accuracy and sensitivity of spectrum annotation	Streamlines the identification of peptides from MS data, making the process faster and more reliable
	Deep learning models predict experimental peptide measurements from amino acid sequences alone [66]	Improves the quality and reliability of analytical workflows	Identifies disease-related biomarkers from proteomics data
Metabolomics	Metabolic Network Reconstruction [67] involves ML	This approach allows for the rapid and accurate mapping of metabolic pathways	It helps in understanding of cellular metabolism and identifying potential targets for metabolic engineering.
	Systems Metabolic Engineering [68] involves ML	Predicts the behavior of complex biological systems under different conditions	Helps design more efficient metabolic pathways and optimize production processes in biotechnology
Drug Discovery	Target Identification/Prioritization [69] The Open Targets Platform uses ML to integrate public domain data, enabling faster and more accurate identification of drug targets	This reduces the time required for target discovery from years to days	Accelerates therapeutics development
	Protein Structure Prediction [70]: AI model—AlphaFold has revolutionized the prediction of protein structures	This reduces the time required from months and years to seconds	Provides crucial insights into how drugs can interact with their targets

## Data Availability

All other data generated or analyzed during this study are included in this published article.

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
