# Peer review of "Drug Discovery in the Age of Artificial Intelligence: Transformative Target-Based Approaches"

_ijms, 2024, doi:10.3390/ijms252212233_

Round 1
Reviewer 1 Report
Comments and Suggestions for Authors
Manuscript "Drug Discovery in the Age of AI: Transformative Target-based 2 Approaches" by Patne et al.
This review describes the different approaches to artificial intelligence and their application to drug discovery. It is well-written and exhaustive, covering the methods and their classifications according to the supervision and the drug design approach. However, the authors did not describe all technical terms, for example, in this paragraph:
"2.1.4.1. Clustering algorithms:
Clustering is suitable for identifying patterns by grouping similar data naturally without drastically effecting the data size. This can be useful for input data which can be difficult to interpret when all the datapoints are not present such as part of input sequence of parts of structural fingerprint ensembles. There are three frameworks for clustering models, k-means, bisecting k-means, and Ward’s algorithm."
The three clustering models were not described. A future reader may be an expert and know all the terminology, but a full description should be helpful for a general reader. As this example, there are several cases throughout the manuscript. Also, some applications of the methods can be described in depth.
Author Response
Comment #1. The reviewer noted the absence of detailed descriptions for technical terms such as clustering models and suggested that more in-depth descriptions be provided for a general readership. They also mentioned other areas across the manuscript where similar clarification would be helpful.
Response:
Thank you for your valuable feedback. We have revised Section 2.1.4.1 by providing descriptions for each clustering model (k-means, bisecting k-means, and Ward’s algorithm) and their specific applications in drug discovery. This addition offers clearer guidance on these terms for readers unfamiliar with clustering models. We have also reviewed the manuscript for similar technical terms and ensured that each is adequately explained to enhance readability and comprehension.
Reviewer 2 Report
Comments and Suggestions for Authors
In this review state of the art Machine learning algorithms applied to the different fields of computational drug discovery have been extensively summirized. The review is outstanding written and organized, although it is hard to understand to reserachers not experienced in the AI filed. I have only a few suggestion to authors:
-a few typos are to be addressed
- tables could better spaced as sometimes the sentences in adjacent columns are too close each other
-in the conclusions section the authors could add a personal consideration about the real possibility to use, now or in the future, ML models for drug discovery without the need of experimental validation.
Author Response
Comment #1: Reviewer 2 suggested addressing minor typos throughout the manuscript.
Response #1:
#We appreciate your attention to detail. We have carefully reviewed the manuscript and corrected all identified typos to improve clarity and readability.
Comment #2: The reviewer suggested improving the spacing in tables, as the proximity of sentences in adjacent columns made them difficult to read.
Response #2:
Thank you for this suggestion. We have reformatted all tables to ensure that the spacing between columns provides clear separation, making them more legible and improving the overall layout.
Comment #3: In the conclusion, the reviewer recommended adding a personal perspective on the potential of machine learning models in drug discovery without experimental validation.
Response #3:
We have expanded the conclusion section to include our perspective on the feasibility of using machine learning models as standalone tools in drug discovery. This addition explores the current limitations of relying solely on computational predictions and underscores the complementary role of experimental validation in ensuring reliability. We also discuss the future potential of these models as their accuracy and reliability continue to advance.